EMBO
Molecular Medicine

# Deletion of *F4L* (ribonucleotide reductase) in vaccinia virus produces a selective oncolytic virus and promotes anti-tumor immunity with superior safety in bladder cancer models

Kyle G Potts[1,2,3], Chad R Irwin[2,3,4], Nicole A Favis[2,3,4], Desmond B Pink[1,3], Krista M Vincent[1,3,5], John D Lewis[1,3] (iD), Ronald B Moore[1,3,6], Mary M Hitt[1,2,3,†] & David H Evans[2,3,4,†,*] (iD)

## Abstract

Bladder cancer has a recurrence rate of up to 80% and many patients require multiple treatments that often fail, eventually leading to disease progression. In particular, standard of care for high-grade disease, Bacillus Calmette–Guérin (BCG), fails in 30% of patients. We have generated a novel oncolytic vaccinia virus (VACV) by mutating the *F4L* gene that encodes the virus homolog of the cell-cycle-regulated small subunit of ribonucleotide reductase (RRM2). The *F4L*-deleted VACVs are highly attenuated in normal tissues, and since cancer cells commonly express elevated RRM2 levels, have tumor-selective replication and cell killing. These *F4L*-deleted VACVs replicated selectively in immune-competent rat AY-27 and xenografted human RT112-luc orthotopic bladder cancer models, causing significant tumor regression or complete ablation with no toxicity. It was also observed that rats cured of AY-27 tumors by VACV treatment developed anti-tumor immunity as evidenced by tumor rejection upon challenge and by *ex vivo* cytotoxic T-lymphocyte assays. Finally, *F4L*-deleted VACVs replicated in primary human bladder cancer explants. Our findings demonstrate the enhanced safety and selectivity of *F4L*-deleted VACVs, with application as a promising therapy for patients with BCG-refractory cancers and immune dysregulation.

**Keywords** bladder cancer; immunotherapy; oncolytic virus; ribonucleotide reductase; vaccinia virus
**Subject Categories** Cancer; Genetics, Gene Therapy & Genetic Disease; Urogenital System

## Introduction

Over 90% of cases of bladder cancer are subtyped as urothelial cell carcinomas. When first diagnosed, about 80% of these cases are classified as non-muscle-invasive bladder cancer (NMIBC) [reviewed in (Anastasiadis & de Reijke, 2012; Potts *et al*, 2012; Delwar *et al*, 2016)], but unfortunately up to 80% of these patients will experience a recurrence within 5 years of initial treatment (van Rhijn *et al*, 2009). Current standards of care for low-risk patients include surgery and intravesical chemotherapy (Shen *et al*, 2008). The high-grade (Ta, T1, or carcinoma *in situ*) tumors are most likely to recur, and treatment for these patients includes surgery often followed by intravesical therapy with Bacillus Calmette–Guérin (BCG) (Shen *et al*, 2008). The side effects of BCG treatment include a risk of infection, cystitis, and prostatitis, and it can be hazardous for immunocompromised patients (Lamm *et al*, 1992). Moreover, about 30% of patients fail BCG therapy leaving cystectomy as the next common treatment option (Zlotta *et al*, 2009). How BCG works is poorly understood and it may simply be a pro-inflammatory agent (Redelman-Sidi *et al*, 2014). There is no evidence to suggest that BCG generates protective anti-tumor immunity and this may partly explain the high rate of treatment failure (Biot *et al*, 2012). In fact, there has been very little improvement in the treatment of high-grade NMIBC in the last 10–20 years and recurrence after BCG therapy is still one of the most significant problems in the management of bladder cancer (Downs *et al*, 2015). This highlights the urgent need for safer and more reliable bladder-sparing approaches.

Oncolytic viruses are intended to replicate selectively in, and kill, cancer cells while sparing normal tissues [reviewed in (Potts *et al*, 2012; Russell *et al*, 2012; Kaufman *et al*, 2015)]. Some of the many cell pathways that affect virus replication are those that regulate cell

1 Department of Oncology, Faculty of Medicine and Dentistry, University of Alberta, Edmonton, AB, Canada
2 Li Ka Shing Institute of Virology, Faculty of Medicine and Dentistry, University of Alberta, Edmonton, AB, Canada
3 Cancer Research Institute of Northern Alberta (CRINA), Faculty of Medicine and Dentistry, University of Alberta, Edmonton, AB, Canada
4 Department of Medical Microbiology & Immunology, Faculty of Medicine and Dentistry, University of Alberta, Edmonton, AB, Canada
5 Department of Anatomy & Cell Biology, Faculty of Medicine and Dentistry, University of Western Ontario, London, ON, Canada
6 Department of Surgery, Faculty of Medicine and Dentistry, University of Alberta, Edmonton, AB, Canada
*Corresponding author. Tel: +1 780 492-7997; E-mail: devans@ualberta.ca
†These authors contributed equally to this work

   

proliferation and DNA replication, processes that are critically dependent upon deoxynucleoside triphosphate (dNTP) production (Aye et al, 2015). The rate-limiting step in dNTP biosynthesis is the de novo reduction of ribonucleoside diphosphates (rNDPs) to deoxyribonucleoside diphosphates (dNDPs) by the enzyme ribonucleoside diphosphate reductase (RNR) (Nordlund & Reichard, 2006). Since DNA virus replication requires dNTPs, this requirement for RNR activity provides an important biological feature that can be used to target DNA viruses to cancer cells.

Many viruses exhibit oncolytic properties and a modified herpesvirus, Talimogene laherparepvec (T-Vec), recently received US clinical approval (Andtbacka et al, 2015). Vaccinia virus (VACV) has also been studied extensively as an oncolytic agent, with JX-594 (Pexa-Vec) having completed multiple phase II trials (Hwang et al, 2011; Heo et al, 2013; Cripe et al, 2015; Park et al, 2015). VACV is a large double-stranded DNA virus that efficiently infects many different cell types and encodes many of the proteins required for robust replication in normal cells (McFadden, 2005). These proteins include thymidine kinase (TK; the J2R gene product) and both large (RRM1; I4L gene product) and small (RRM2; F4L gene product) subunits of the heterodimeric RNR complex (Slabaugh et al, 1988). The virus-encoded components of RNR complex with each other and can form chimeras with cellular homologs (Hendricks & Mathews, 1998; Gammon et al, 2010). Most oncolytic VACVs reported to date encode mutations in J2R and little research has been conducted to determine whether mutating the RNR genes (Fend et al, 2016) might also produce advantageous oncolytic properties. The F4L gene is an important determinant of VACV virulence and viruses lacking F4L (ΔF4L) are attenuated in vivo whereas ΔI4L mutants are not (Gammon et al, 2010). The fact that cellular RRM2 is cell-cycle-regulated whereas RRM1 is constitutively expressed can perhaps explain this observation (Eriksson et al, 1984) and leads to the prediction that a ΔF4L virus should replicate selectively in dividing cancer cells. This complementation-based strategy might be especially useful for treating more aggressive bladder cancers since increased levels of cellular RRM2 predict a poorer prognosis (Morikawa et al, 2010).

Here we describe pre-clinical studies showing that VACV can be used safely as an intravesical treatment for NMIBC. We find that F4L-deleted VACVs retain much of their cytotoxicity and replication proficiency in bladder cancer cells. F4L-deleted VACVs also safely and effectively clear bladder tumors in animal models and induce a durable anti-tumor immunity. These findings highlight the potential for using a F4L-deleted VACV in treating bladder cancer, especially in patients who have failed BCG treatment or are immunosuppressed.

# Results

### Growth of VACV ΔF4L and ΔJ2R mutants in vitro

Homologous recombination was used to disrupt the VACV (strain Western Reserve) F4L and J2R loci as shown in Fig EV1. Thirteen out of fifteen bladder cancer cell lines grown under high serum conditions (10%) supported robust virus replication, exceptions being UM-UC3-luc and UM-UC9 cells (Figs 1A and EV2). Under low serum conditions (0.1%), the wild-type (WT) and ΔJ2R VACV grew as well as was seen in 10% serum. Although viruses lacking F4L

also replicated efficiently in most of the cancer cell lines under low serum conditions, the ΔF4L VACVs grew poorly in 253J and AY-27 cells (Fig 1B). Most importantly, compared to WT, growth of ΔF4LΔJ2R and ΔF4L VACVs in low serum was reduced > 4,000-fold in the NKC (normal epithelial kidney) line and > 250-fold in N60 (normal fibroblast) cell line, whereas the growth of ΔJ2R VACV was only marginally reduced compared to WT in the NKC and N60 cells under the same low serum conditions (Figs 1 and EV2).

The effect of VACV on cell survival was determined using a resazurin-based viability assay. Similar to growth of virus under high serum conditions (Fig 1A), no dramatic difference in the efficiency of virus-mediated cell killing was seen among the different viruses under high serum conditions (Figs 1C and EV3). However, under low serum conditions, both N60 normal skin fibroblasts and NKC epithelial kidney cells became relatively resistant to ΔF4L and ΔF4LΔJ2R VACV killing. Interestingly, in low serum conditions, 253J and AY-27 cancer cells were still highly susceptible to killing by ΔF4LΔJ2R VACV (Fig 1D), even though virus replication was attenuated. This was a specific property of the ΔF4LΔJ2R virus; 253J and AY-27 cells were still relatively resistant to the ΔF4L VACV. Finally, both N60 and NKC cells grown in 0.1% fetal bovine serum (FBS) showed a low proportion of cells in S-phase whereas the proportion of RT112-luc cells in S-phase remained high (Fig EV4), suggesting that proliferation status under our low serum growth conditions may mimic the proliferation status of normal and tumor tissues in vivo. These data indicate that the mutant VACVs, in particular ΔF4LΔJ2R VACV, retained much of the cytotoxic capabilities and replication proficiency of WT virus in bladder cancer cells but do not replicate in non-dividing cells.

### Nucleotide biosynthetic proteins are elevated in bladder cancer cells

One might expect that ΔF4L and/or ΔJ2R strains would depend upon complementation from cellular RRM2 and TK1, respectively, to provide the dNTPs required for virus replication. Some limited data suggested that ΔF4L VACVs do grow better in cells expressing higher levels of RRM2 (Gammon et al, 2010). To examine this matter in more detail, the levels of proteins catalyzing nucleotide biosynthesis were quantified in a panel of human bladder cancer cell lines and in normal N60 fibroblasts under 10 and 0.1% serum conditions (Fig 2A and Appendix Fig S1). Western blots showed a general elevation in the levels of cellular RRM1, RRM2, and TK1 in cancer cell lines compared to normal cells (Appendix Fig S1). The abundance of the DNA damage-inducible form of R2, p53R2, did not significantly differ between the cancer cell lines and in normal skin fibroblasts.

To demonstrate F4L-deleted VACVs' dependence on cellular RRM2, we tested whether knockdown of RRM2 in HeLa cells would prevent VACV replication. Efficient knockdown was achieved following transfection with RRM2-specific siRNAs, as confirmed by Western blot analysis (Fig 2B). The cells were then infected with the different VACVs and virus yield measured by plaque assay (Fig 2C). There was a significant reduction in ΔF4L and ΔF4LΔJ2R VACV replication in cells with RRM2 knockdown, while WT or ΔJ2R VACV replication was unaffected.

We also examined expression of nucleotide metabolism proteins in samples isolated from primary human bladder tumors and from normal bladder urothelium. Western blot analysis showed elevated

   

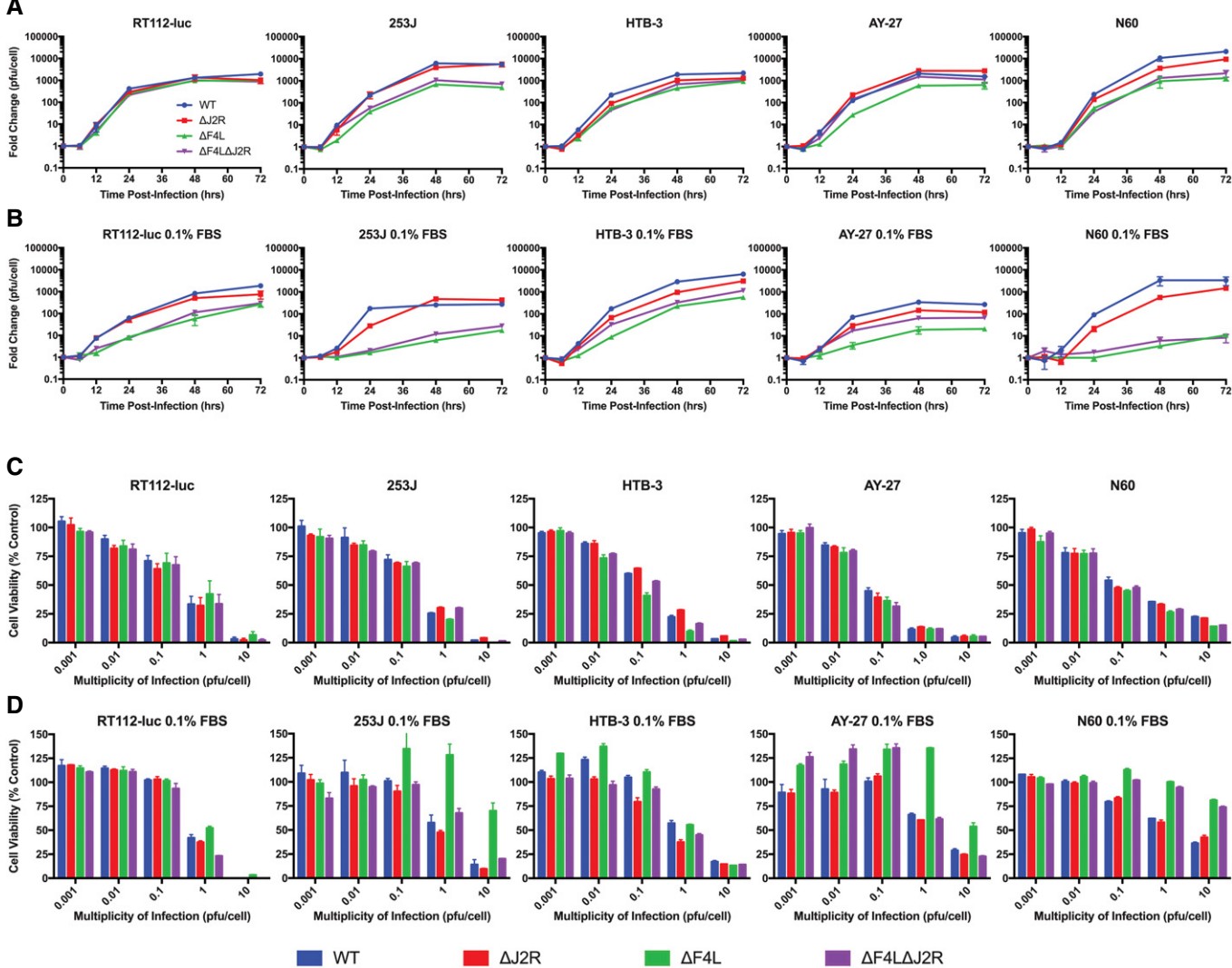

**Figure 1.   ΔF4LΔJ2R VACV retains much of the replication proficiency and cytotoxicity of WT VACV in bladder cancer cells.**

A, B   Growth curves for the indicated VACV strains in subconfluent human bladder cancer cell lines, a rat bladder cancer cell line (AY-27), and a normal human skin fibroblast line (N60). The cells were infected with 0.03 PFU/cell. (A) Panel of cells grown under normal serum conditions. (B) Panel of cells grown under low (0.1%) serum conditions. Cultures were harvested at the indicated times and titered on BSC-40 cells.

C, D   Survival of cell lines infected *in vitro* with the indicated VACV strains. Subconfluent cells were infected at the indicated multiplicities of infection (in PFU/cell). Uninfected cells were used as control. (C) Panel of cells grown under normal serum conditions. (D) Panel of cells grown under low (0.1%) serum conditions. The cells were incubated with resazurin to assess viability 3 days post-infection relative to uninfected control cells.

Data information: Mean ± SEM is shown. For (A) and (B), data represent at least two independent lysates titered in duplicate. For (C) and (D) $n \geq 3$.

expression of both RRM1 and RRM2 in the tumor tissues relative to the normal urothelium (Fig 2D). Additionally, TK1 was only detectable in the tumor lysates, with two of these showing high expression levels, and only minimal expression in the remaining lysates. As in cultured cells, p53R2 expression was not specifically associated with tumors. These observations were generally corroborated by gene expression data obtained from primary tumor samples previously analyzed by Sanchez-Carbayo *et al* (2006). Reanalysis of these data showed that *RRM2* and *TK1* expression were significantly increased in both NMIBC and muscle-invasive bladder cancers (MIBC) when compared to the normal urothelium (Fig 2E). In contrast, *RRM1* was only significantly over-expressed in MIBC.

The expression level of these same proteins was also measured in different tissues recovered from an orthotopic rat AY-27 bladder cancer model (Fig 2F). We detected high RRM2 expression in tumor tissue as well as normal bladder. RRM1 did not appear elevated in tumors compared to normal tissues, and very little TK1 expression was detected in any of the tissues.

**VACVs encoding *F4L* and *J2R* mutations safely clear human bladder tumor xenografts**

The safety and oncolytic activity of the mutant VACVs was tested in xenograft models of human bladder cancer. These models were

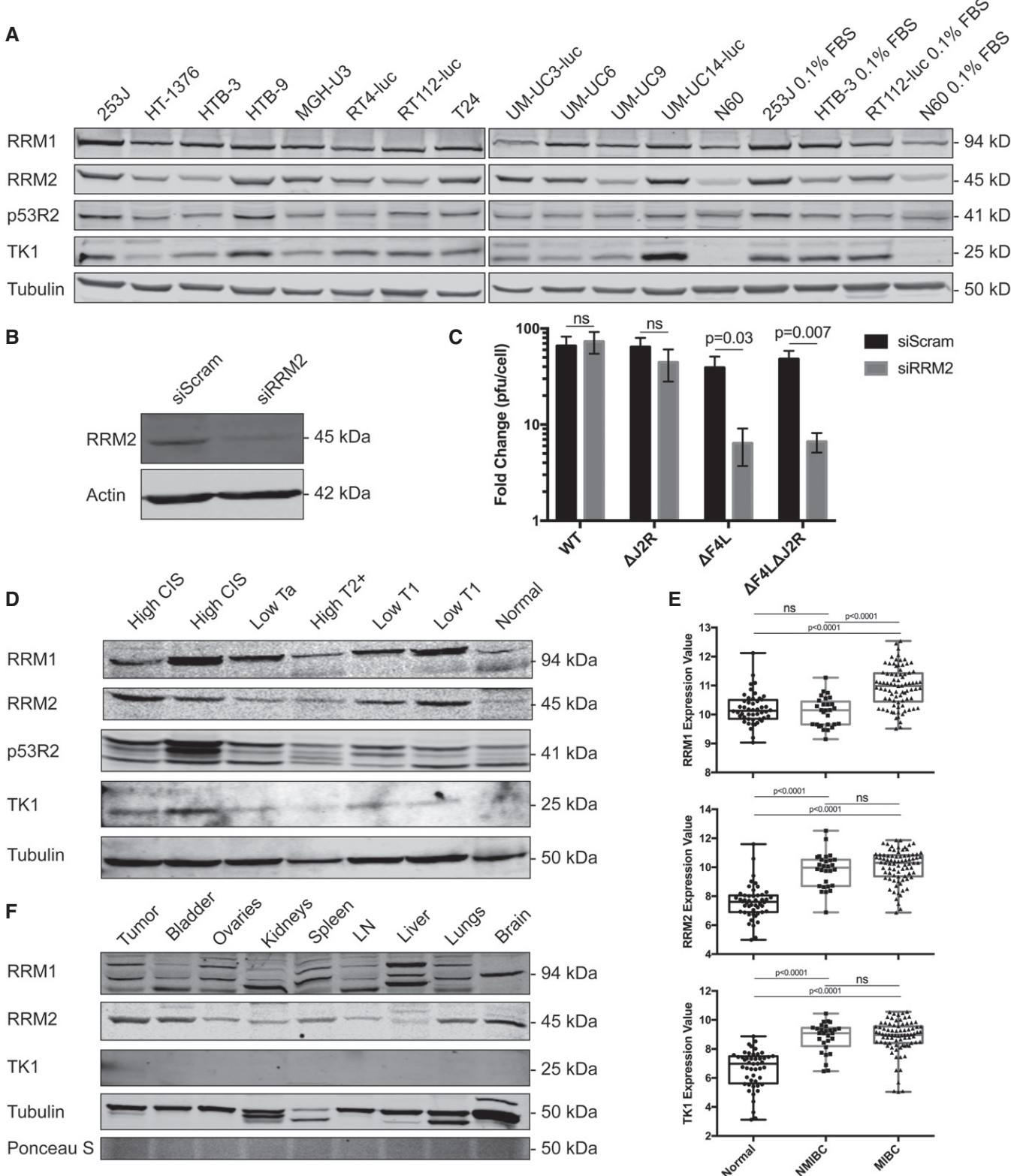

**Figure 2.**

established by subcutaneous or orthotopic implantation of luci-ferase-expressing human RT112 cells (RT112-luc) in Balb/c immune-deficient mice. In the first study, we injected three doses of virus, each comprising $10^6$ PFU of ΔJ2R, ΔF4L, ΔF4LΔJ2R, or UV-inactivated VACV as a control, directly into subcutaneous RT112-luc tumors (Fig 3A). An mCherry signal, indicative of virus replication,

**Figure 2.  Elevated levels of proteins catalyzing nucleotide biosynthesis in bladder cancer cell lines and primary human tumor lysates.**

A   Western blot showing RRM1, RRM2, p53R2, and TK1 expression in human bladder cancer cell lines and N60 normal human fibroblasts. β-tubulin is shown as a loading control.

B   siRNA depletion of RRM2 in HeLa cells 3 days post-transfection as determined by Western blot analysis.

C   Growth of the indicated VACV strains in subconfluent HeLa cells. The cells were treated for 24 h with a scrambled control siRNA ("Scram") or an *RRM2*-targeted siRNA and then infected with the indicated viruses at 0.03 PFU/cell. The cultures were harvested 2 days later and titered on BSC-40 cells.

D   Western blot showing RRM1, RRM2, p53R2, and TK1 expression levels in human primary tumor tissues and adjacent normal urothelium. β-tubulin is shown as a loading control.

E   Analysis of *RRM1*, *RRM2*, and *TK1* expression levels from publicly available patient bladder cancer microarray data (NMIBC: non-muscle-invasive bladder cancer; MIBC: muscle-invasive bladder cancer). Data points denote $\log_2$-transformed MAS5.0 normalized values. The box limits represent the upper and lower quartiles. The median is marked by the horizontal line inside the box. The whiskers extend to the highest and lowest observed values.

F   Western blot showing RRM1, RRM2, and TK1 expression in rat AY-27 bladder tumor tissue and the indicated normal tissues. β-tubulin and Ponceau S staining are shown as loading controls. In all Western blots, equal amounts of total protein (30 μg) were assayed.

Data information: Mean ± SEM is shown. For (C) $n = 4$ and significance was determined by multiple *t*-test. Microarray data in (E) were analyzed using RStudio (v0.98.501) and significance analysis was performed using a one-way ANOVA followed by Tukey's HSD. Western blots are representative of at least two or three independent experiments.

was detected in all mice before the third live virus injection (Appendix Fig S2) and all animals treated with live virus showed significantly prolonged survival compared to those treated with UV-inactivated VACV (Fig 3B). Both ΔF4L and ΔF4LΔJ2R VACV significantly increased survival compared to animals treated with the ΔJ2R strain ($P = 0.015$ and $P = 0.001$, respectively). Tumor growth was controlled in all animals treated with live viruses as determined by caliper measurements (Fig 3C), and by luciferase detection (Fig 3F and G, and Appendix Fig S3).

The ΔJ2R virus showed strong anti-tumor activity, but this was only achieved with significant toxicity in Balb/c immune-deficient mice. Seven of ten ΔJ2R VACV-treated mice were euthanized due to excessive weight loss (Fig 3D). The mice euthanized prior to day 50 exhibited significant viremia at endpoint (Fig 3E) whereas the mice euthanized at later time points had acquired systemic *Staphylococcus aureus* infections. These, as well as other ΔJ2R VACV-treated mice, exhibited transient and spontaneously resolving dermal pox lesions at sites distal to the tumor injection site, which may have provided a route for the bacterial infection. In contrast, all mice treated with ΔF4LΔJ2R VACV were completely cured of their RT112-luc tumors and continued to gain weight throughout the experiment (Fig 3D). We saw no signs of toxicity or virus lesions in this treatment group. A pilot experiment using a subcutaneous UM-UC3-luc xenograft model also suggested strong oncolytic activity from ΔF4LΔJ2R VACV (Appendix Fig S4) even though UM-UC3-luc cells supported only limited VACV growth *in vitro* (Fig EV2).

We next developed a new orthotopic RT112-luc xenograft model to replace the KU7 model that was recently shown to have been contaminated with HeLa cells (Jäger *et al*, 2013). The treatment scheme is shown in Fig 4A. Bioluminescence images show a continuous increase in luciferase signal from tumors treated with UV-inactivated virus, and a decline in signal from all live-VACV-treated animals, with most tumors eventually being cleared (Fig 4B–D). To measure virus distribution after intravesical treatment, we euthanized mice 3 days after the last virus instillation and measured virus titers in tumors and other organs. We detected little or no spread of the ΔF4L or ΔF4LΔJ2R virus to other organs (Fig 4F). In one animal, significant levels of ΔF4LΔJ2R VACV were detected in the kidney, but this coincided with a luciferase signal indicating tumor spread to this site, demonstrating the tumor selectivity of this virus. In contrast, ΔJ2R VACV was detected in several organs in treated mice (Fig 4F). Delivering the virus directly into the bladder caused no

toxicity as judged by animal weights (Fig 4E). However, as was seen in the subcutaneous model, most of the ΔJ2R-treated mice developed pox lesions on their backs (Fig 4G).

Finally, we tested efficacy and toxicity when virus was administered by intravenous injection in a subcutaneous RT112-luc xenograft tumor model. ΔJ2R VACV replication was detectable at the tumor site by day 20 (Appendix Fig S5, but this virus caused significant toxicity and showed no significant survival benefit over UV-inactivated virus by log-rank test (Appendix Fig S6A–D). In contrast, there was a significant increase in the survival of mice treated with the ΔF4L ($P = 0.0015$) and ΔF4LΔJ2R ($P = 0.013$) viruses (Appendix Fig S6A and B). The ΔF4LΔJ2R group exhibited virus-encoded mCherry fluorescence at the tumor site by day 27 (Appendix Fig S6), while this signal was not seen until day 40 in the ΔF4L group.

### VACV mutants clear syngeneic orthotopic rat bladder tumors

The mutant VACVs were also tested using an orthotopic immune-competent AY-27 rat bladder cancer model. AY-27 tumors resemble high-grade urothelial cell carcinoma in both morphology and tumor biology, providing an excellent model of human bladder cancer (Xiao *et al*, 1999). Animals were treated by intravesical instillation with $3 \times 10^8$ PFU of live ΔJ2R, ΔF4L, or ΔF4LΔJ2R VACV, or UV-inactivated virus (Fig 5A). By day 35, there was a significant reduction in the growth rate of all tumors treated with live virus (Fig 5B–D). Following ΔF4LΔJ2R VACV treatment, representative cystoscopic images of the rat bladders revealed tumor necrosis and tumor elimination with little or no inflammation in normal urothelium (Fig 5C). On day 15 after tumor implantation (3 days after final virus instillation), ΔF4L and ΔF4LΔJ2R VACVs could only be detected in the tumor, whereas ΔJ2R VACV was found in the tumor and in the ovaries, kidneys, and lungs (Fig 5E). Of particular concern was the apparent development of cysts on the ovaries of some rats (Fig 5F). However, none of the animals treated with any virus showed signs of overt toxicity. The most important result of this study is that all live VACV treatments significantly increased survival ($P < 0.001$) when compared to the UV-inactivated control (Fig 5G). It is noteworthy that although nine of the ΔJ2R VACV-treated animals appeared to be tumor-free by day 75, three of these rats later developed rapidly growing recurrent tumors. This was not seen with either ΔF4L or ΔF4LΔJ2R VACV as far out as 125 days into the study. Interestingly, we also observed significantly higher levels

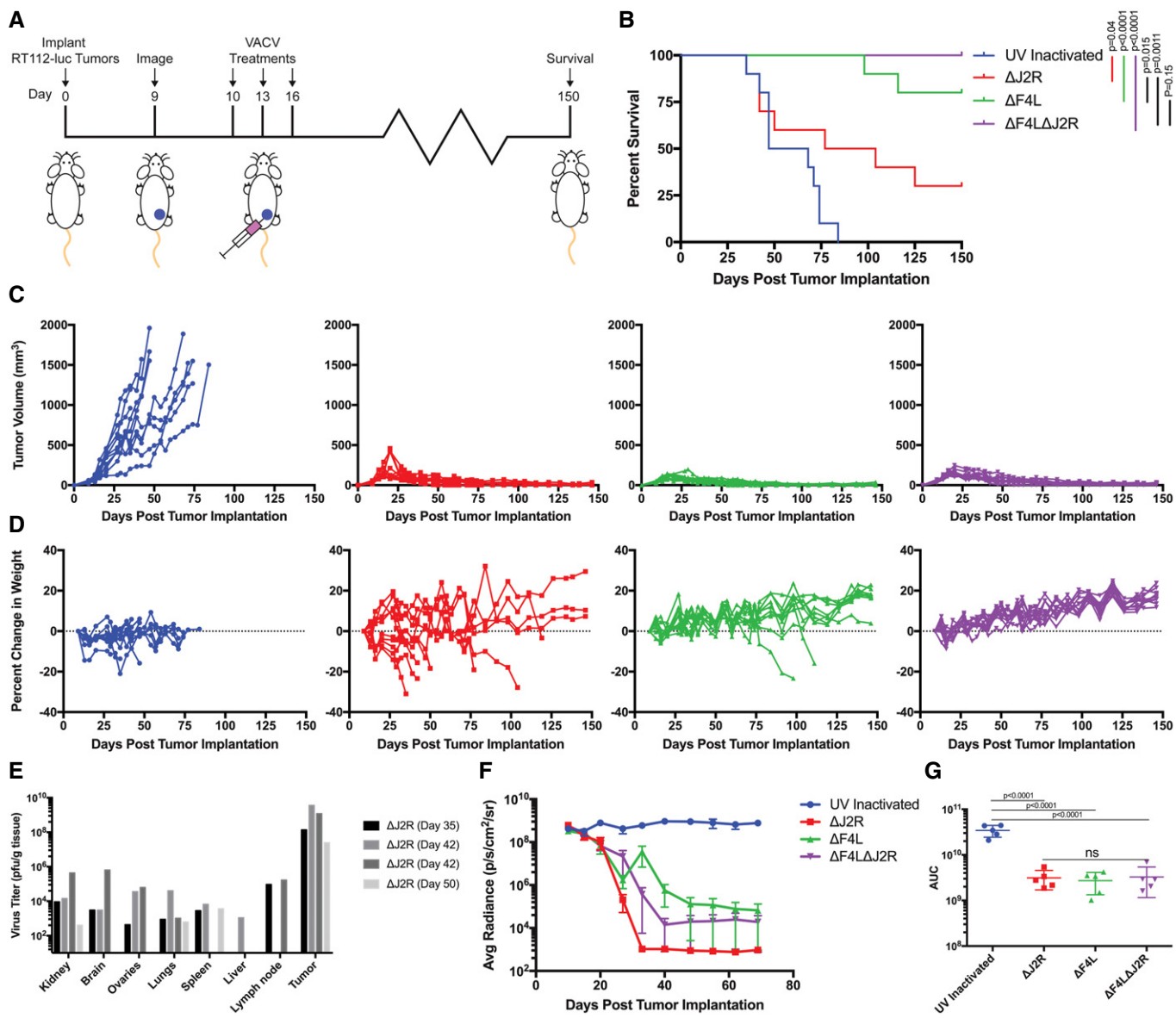

**Figure 3.  ΔF4LΔJ2R VACV safely and effectively clears subcutaneous human RT112-luc xenografted tumors.**

A   Experimental scheme. Balb/c nude mice were injected with $2 \times 10^6$ RT112-luc cells in the left flank at day zero. Then, $10^6$ PFU of UV-inactivated, ΔJ2R, ΔF4L, or ΔF4LΔJ2R VACV were injected into the tumors on days 10, 13, and 16 post-implantation.

B   Overall survival of immunocompromised mice bearing RT112-luc flank tumors following treatment with the indicated viruses (n = 10 per group).

C   Growth of individual virus-treated RT112-luc tumors. Legend as in (B).

D   Analysis of individual animal body weights plotted as mean change in body weight relative to day 10. Legend as in (B).

E   VACV titers in tissues taken from animals euthanized due to toxicity (note: only mice that had detectable (4/10) virus as determined by plaque assay are shown).

F   Quantification of average luminescence (an indication of live tumor cells) from bladder tumors corresponding to (B).

G   Area under the curve (AUC) calculation from the data in (F) (n = 5 per group).

Data information: Mean ± SEM is shown. Animal survival was analyzed by log-rank (Mantel–Cox) test in (B). One-way ANOVA followed by Tukey's multiple comparison test was used in (G). For luciferase quantification in (F) and AUG calculations in (G) n = 5 representative animals.

of anti-VACV antibodies in animals treated with ΔJ2R VACV relative to the F4L-deleted VACVs (Fig 6A).

## Cured animals develop protective anti-tumor immunity

To test whether animals with complete tumor responses had developed anti-tumor immunity, we implanted fresh AY-27 cells in the bladders of eleven surviving VACV-treated animals and all cured animals rejected tumor implantation (Appendix Fig S7). Systemic anti-tumor immunity was also tested by implanting fresh AY-27 tumor cells subcutaneously in the flanks of six cured ΔF4LΔJ2R-treated animals. Again, all cured rats were protected from tumor development whereas significant tumor growth (P < 0.001) was seen in the age-matched controls (Fig 6B).

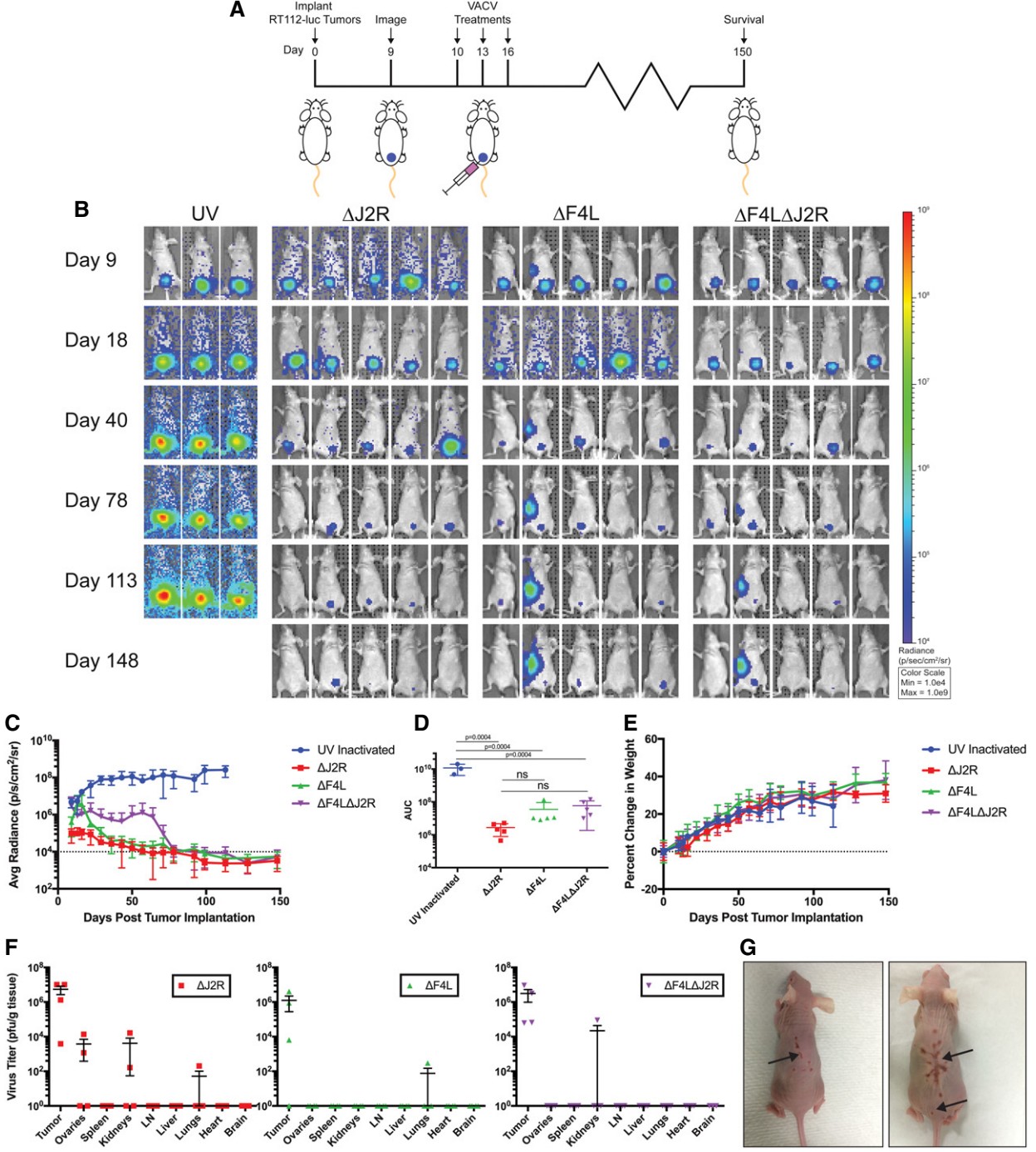

**Figure 4. ΔF4LΔJ2R VACV safely and effectively clears orthotopic human RT112-luc xenografted tumors.**

A   Experimental scheme. Balb/c nude mice were instilled with $2 \times 10^6$ RT112-luc cells on day zero. Mice were imaged for luciferase following luciferin injection on day 9 to verify tumor implantation. On each of days 10, 13, and 16 post-tumor implantations, $10^6$ PFU of UV-inactivated, ΔJ2R, ΔF4L, or ΔF4LΔJ2R VACV was instilled into the bladder and left in-dwell for 1 h. N = 5 per group.

B   Representative luminescence images from animals bearing orthotopic RT112-luc tumors and treated with VACVs.

C   Quantification of average luminescence, the dashed line indicates limit of detection.

D   Area under the curve calculation from the data in (C).

E   Analysis of individual animal body weights plotted as mean change in body weight.

F   Virus titers in tissues on day 19. Organs were harvested, homogenized, and the virus was titered on BSC-40 cells with n = 4 mice per group.

G   Representative images of lesions on two mice taken approximately 125 days post-tumor implantation. Arrows indicate lesions.

Data information: Mean ± SEM is shown. For (C), (D), and (E) n = 3 for UV-inactivated and 5 for live viruses. One-way ANOVA followed by Tukey's multiple comparison test was used in (D).

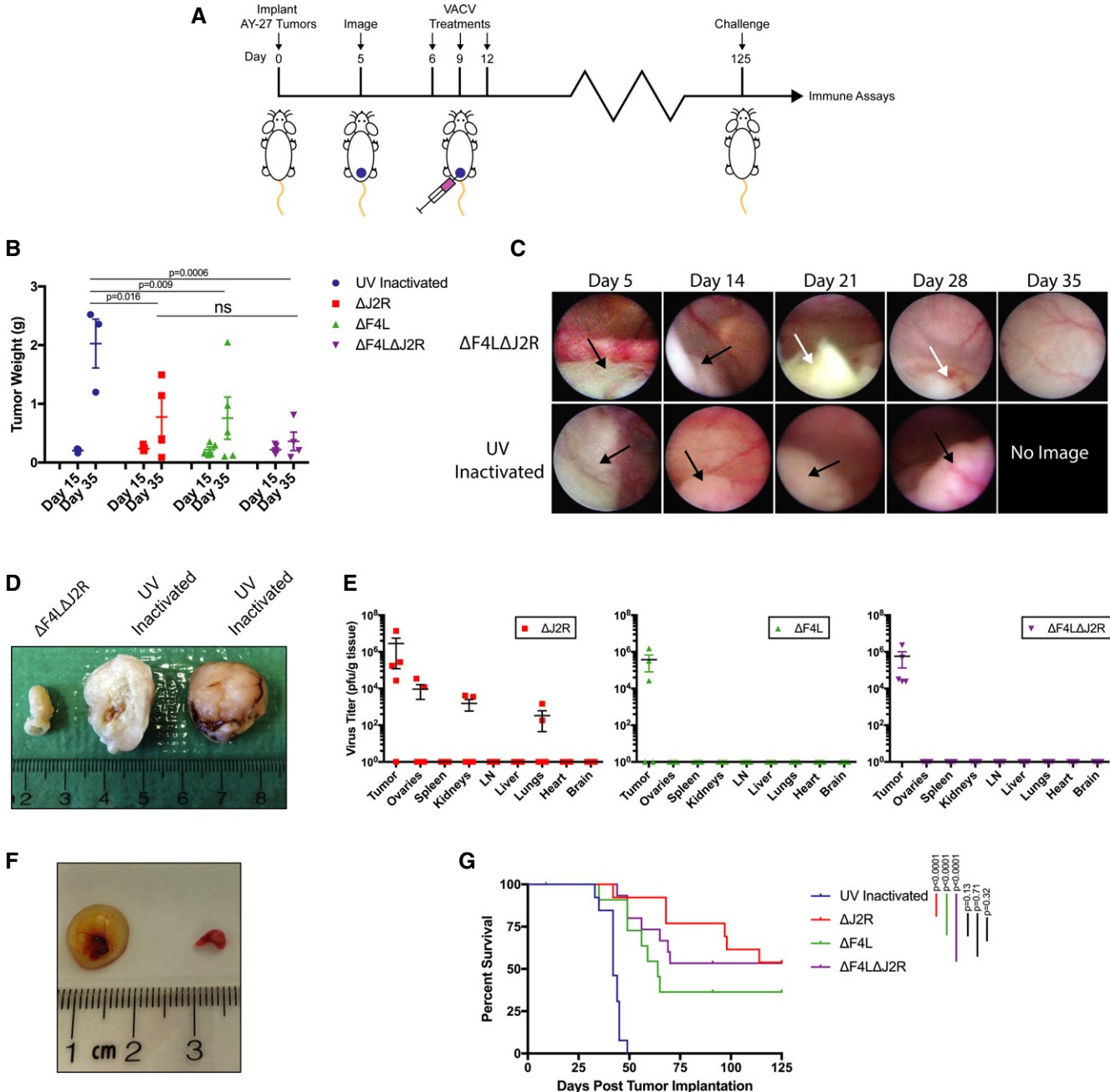

**Figure 5.  ΔF4LΔJ2R VACV safely clears rat orthotopic AY-27 syngeneic tumors and generates protective anti-tumor immunity.**

A  Experimental scheme. Rats were instilled in the bladder with $3 \times 10^6$ AY-27 cells on day zero and cystoscoped on day 5 to verify tumor engraftment. Then, $3 \times 10^8$ PFU of UV-inactivated, ΔJ2R, ΔF4L, or ΔF4LΔJ2R VACV were instilled into the bladder of each rat on each of days 6, 9, and 12.

B  Tumor weight from animals euthanized on days 15 and 35 ($n = 5$ for each day).

C  Representative cystoscope images of the bladders of a ΔF4LΔJ2R virus-treated rat and a UV-inactivated virus-treated rat on days 5, 14, 21, 28, and 35. Black arrow indicates tumor and white arrow indicates necrotic tumor tissues.

D  Images of rat bladders treated with ΔF4LΔJ2R VACV (left image) or UV-inactivated VACV (center and right images) on days 6, 9, and 12, and then excised on day 35. The tumor treated with UV-inactivated virus has been cut in half. The center sample shows the tumor interior; the right sample shows the exterior.

E  Virus titers in tissues taken from animals euthanized on day 15 post-implantations. Organs were harvested and homogenized and then lysates were titered on BSC-40 cells. Data for each organ represent $n = 5$ rats per group.

F  Ovaries from rats euthanized 15 days post-tumor implantations and 3 days following final treatment with ΔJ2R VACV (left) or ΔF4LΔJ2R VACV (right).

G  Overall survival of immunocompetent rats bearing AY-27 bladder tumors following treatment with the indicated VACVs. Data represent combined survival of two independent experiments ($n = 12–15$).

Data information: Mean ± SEM is shown. Two-way ANOVA followed by Tukey's multiple comparison test was used in (B). Animal survival was analyzed by log-rank (Mantel–Cox) test in (G).

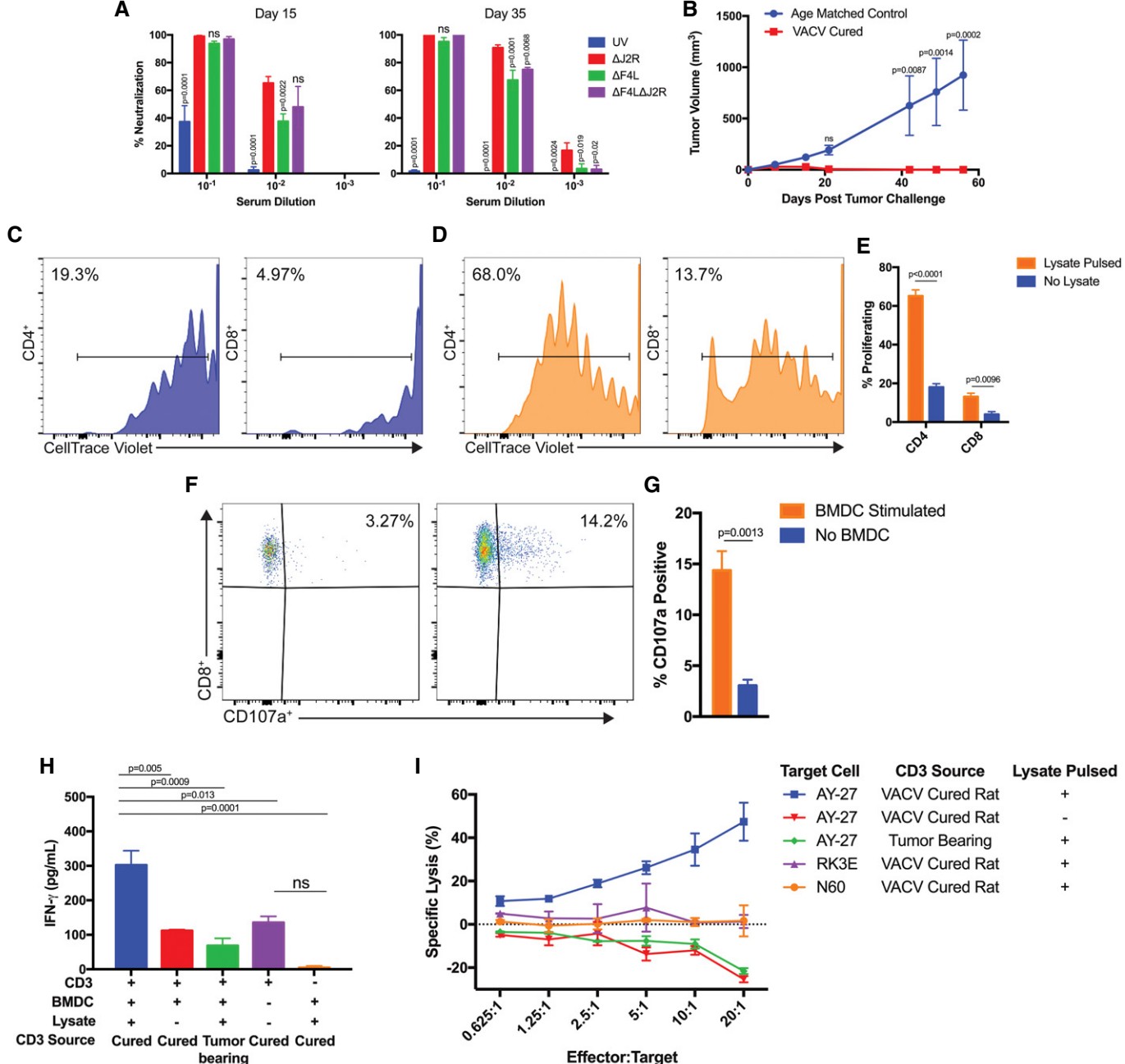

**Figure 6. VACV activates immune responses in rats bearing AY-27 bladder tumors.**

A    VACV-neutralizing antibodies were measured in virus-treated rats 15 and 35 days after implantation ($n$ = 4–5 rats per group).

B    Protection from subcutaneous tumor challenge after virus-induced tumor clearance. AY-27 cells were implanted in the flanks of cured rats ($n$ = 6) and naïve age-matched control rats ($n$ = 4).

C–E    T-cell proliferation after co-culturing with bone marrow-derived dendritic cells (BMDCs). CD4$^+$ and CD8$^+$ cells were co-cultured with BMDCs and proliferation assayed with CellTrace Violet. The representative plots show CD4$^+$ and CD8$^+$ T-cell proliferation after co-culture with either mock-pulsed (C) or with tumor-lysate-pulsed BMDCs (D). Panel (E) shows the percentage of CD4$^+$ and CD8$^+$ T cells that proliferated in response to BMDC stimulation ($n$ = 3).

F, G    *Ex vivo* upregulation of CD107a by CD8$^+$ T cells from challenged rats. (F) CD3$^+$ cells were incubated +/− BMDCs for 1 h in the presence of anti-CD107a antibody, incubated for 5 h with monensin and brefeldin A, and then stained with anti-CD4 and anti-CD8 antibodies. Events were gated for viable CD8$^+$ T cells. Panel (G) shows the percentage of CD107a$^+$ CD8$^+$ T cells +/− BMDC stimulation ($n$ = 3).

H    IFN-γ released after 24-h co-culture of CD3$^+$ cells with BMDCs ($n$ = 3–5).

I    T cells activated *ex vivo* by tumor-lysate-pulsed DCs are cytotoxic. After 6 days of co-culture with BMDC, CD3$^+$ cells were incubated for 18 h with 10,000 target cells and at different effector-to-target ratios. Lysis was determined by LDH assay. RK3E are normal rat kidney cells ($n$ = 2–3 performed in duplicate).

Data information: Mean ± SEM is shown. Two-way ANOVA followed by Tukey's multiple comparison test was used in (A), (B), and (H). For (A), significance was determined against the ΔJ2R group. Two-tailed Student's $t$-test was used in (E) and (G).

The cellular anti-tumor immune response was examined in cured animals that were resistant to tumor challenge. One hundred days after subcutaneous challenge with AY-27 cells, cured ΔF4LΔJ2R-treated rats were euthanized, the spleens were removed, and CD3$^+$ cells were isolated. The CD3$^+$ cells were stimulated with bone marrow-derived dendritic cells (BMDCs) previously pulsed with AY-27 tumor lysate to enable antigen presentation (Bachleitner-Hofmann et al, 2002). Both splenic CD3$^+$CD4$^+$ and CD3$^+$CD8$^+$ cells proliferated when stimulated by lysate-pulsed BMDCs (Fig 6C–E). To confirm activation of the CD8$^+$ cells, the CD3$^+$CD8$^+$ population was examined for expression of the CD107a marker (Betts et al, 2003). There were significantly more CD8$^+$CD107a$^+$ cells following stimulation with lysate-pulsed BMDCs (Fig 6F and G). These stimulated CD3$^+$ cells also exhibited elevated secretion of IFN-γ (Fig 6H) and killed AY-27 tumor cells, but not normal RK3E (F344 Fischer rat kidney) or N60 fibroblast cells (Fig 6I). Significantly, CD3$^+$ cells from AY-27-tumor-bearing rats that had never been exposed to VACV could not kill AY-27 cells. Collectively, these data show that VACV treatment generated a durable tumor-antigen-specific cytotoxic T-cell response in the AY-27 rat model.

### VACV can replicate in primary tumor cell cultures and in tumor explants *ex vivo*

We also examined whether ΔF4L and/or ΔF4LΔJ2R VACV could replicate in human bladder cancers in either primary cell cultures or tumor explants. Explanted tissues from a low-grade T1 tumor (UCKP-6), a high-grade T2$^+$ tumor (UCKP-4), and normal urothelium were infected with $10^6$ PFU of each VACV, and replication was detected using a virus-encoded mCherry reporter (Fig 7A–D). In the low-grade tumor, the ΔF4LΔJ2R and ΔJ2R VACVs produced nearly identical mCherry signals, while ΔF4L VACV produced a much lower signal (Fig 7A and B). In high-grade tissues, all viruses produced nearly identical mCherry signals (Fig 7C and D). All viruses had minimal fluorescence in normal urothelium (Fig 7A). We used the same low- and high-grade tumor samples to establish monolayer cell cultures (Fig 7E and F). The ΔF4LΔJ2R VACV grew to the same level as ΔJ2R VACV in both low- and high-grade primary tumor cells, whereas the ΔF4L VACV grew more poorly in the low-grade UCKP-6 culture, just as was seen in the tissue explants. Collectively, these data support the hypothesis that the ΔF4LΔJ2R VACV could be used to treat both high- and low-grade bladder cancer.

### VACV can infect and replicate in BCG-resistant bladder cancer cells

Accumulating evidence shows that attachment and internalization of BCG are required for its anti-tumor activity (Zhao et al, 2000; Redelman-Sidi et al, 2013), which creates a problem when tumors become resistant to infection. We were interested to see whether BCG-resistant bladder cancer cells might still be sensitive to VACV infection. We exposed bladder cancer cells in culture to a Pasteur strain of BCG expressing green fluorescent protein, and measured infection by flow cytometry. These studies showed that T24, 253J, and HTB-3 cells are BCG-sensitive while the RT112-luc cells are resistant to BCG uptake (Fig 8A). Nevertheless, all viruses replicated in both BCG-sensitive and BCG-resistant cell lines (Fig 1A). We also

examined BCG uptake in primary cultures prepared from low-grade Ta (UCKP-16) and low-grade T1 (UCKP-17) bladder tumors. Although both primary cultures were resistant to BCG (Fig 8B), they were still killed by VACV (Fig 8D). The cytotoxicity was similar to that seen in established cell lines (Fig 1C) despite the lower levels of replication (Fig 8C). Interestingly, while ΔF4LΔJ2R VACV replicated to slightly lower levels than WT and ΔJ2R viruses (Fig 8C), it is just as effective at killing these two primary bladder cancer samples, and performed better at killing UCKP-16 cells than the ΔF4L VACV. Overall, these results highlight the potential for using oncolytic VACV, particularly the ΔF4LΔJ2R variant, to treat BCG-refractory bladder cancer.

## Discussion

Bladder cancer has not received much attention as a target for clinical trials of oncolytic virotherapy [reviewed in (Potts et al, 2012; Delwar et al, 2016)]. Currently, the most advanced clinical trial is one using CG0070, a conditionally replicating adenovirus. CG0070 has completed a phase I trial (Burke et al, 2012) and is presently being evaluated in patients with advanced NMIBC who have failed BCG and refuse a cystectomy. However, the efficacy of this treatment is yet to be determined. It is interesting to note that in 2001, four patients with MIBC were treated with smallpox vaccine (Dryvax) intravesically before cystectomy. Three out of the four patients remained disease-free after 4 years, which highlights the potential of VACV as a durable treatment for bladder cancer (Gomella et al, 2001). It is difficult to judge whether one could perform such a study today using the WT Dryvax virus, but these intriguing results suggest that a VACV modified to enhance tumor specificity and reduce virulence might offer a superior therapy for bladder cancer.

Most oncolytic VACVs reported to date bear mutations that inactivate *J2R*. *J2R* encodes the viral thymidine kinase, a critical enzyme in the salvage pathway for nucleotide biosynthesis (Russell et al, 2012). Deleting the *J2R* gene has been shown to reduce virulence while allowing VACV replication in dividing cells (Buller et al, 1985). We have shown here that all our bladder cancer cell lines supported replication of ΔJ2R VACV to nearly the same level seen in cells infected with WT VACV. Surprisingly, we could not detect cellular TK1 in the normal N60 cell line even though it supports robust ΔJ2R VACV growth. This could be explained by the fact that thymidine triphosphate can also be produced from deoxyuridine-5′-monophosphate by thymidylate synthase and thymidylate kinase (Romain et al, 2001) and VACV encodes the latter enzyme.

To produce a safer and more tumor-selective oncolytic VACV, we investigated deletion of *F4L*, a homolog of the *RRM2* gene encoding the small subunit of ribonucleotide reductase. We used a panel of bladder cancer cell lines and primary tissues (Fig 2) to confirm reports that RRM2 is elevated in bladder cancer (Morikawa et al, 2010). Cancer cells often undergo metabolic reprogramming because of aberrant oncogenic signaling, adopting a state of anabolic metabolism to generate the needed macromolecules and dNTPs for division (Ward & Thompson, 2012). Additionally, cancer cells generally have an increased S-phase fraction compared to normal cells (Aye et al, 2015). These characteristics of bladder tumor cells may partly explain the

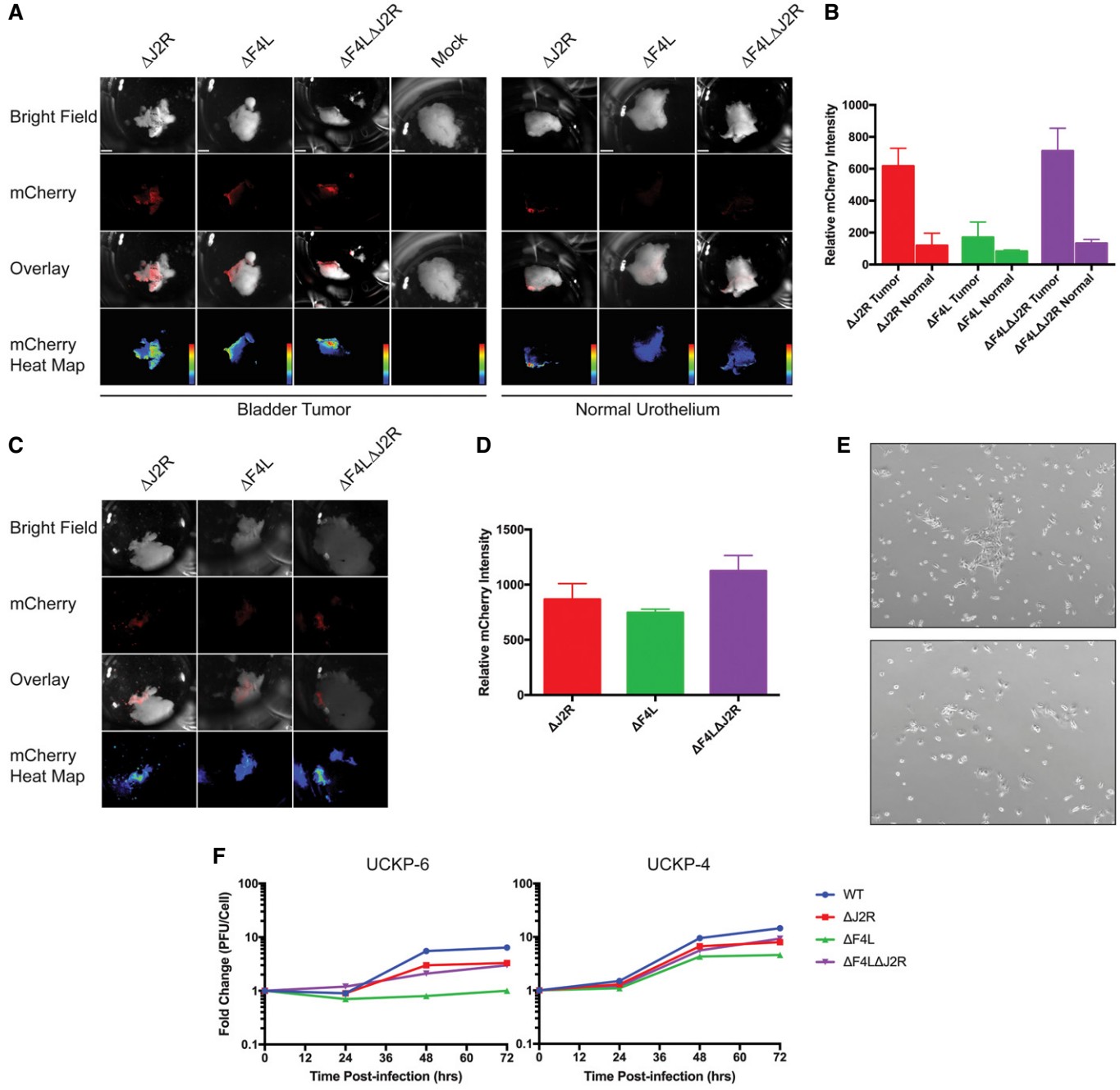

**Figure 7. VACV infects and selectively replicates in primary bladder tumor tissue.**

A   Viruses encoding mCherry fluorescent protein were used to infect primary low-grade T1 (UCKP-6) and normal urothelial tissue samples from patients undergoing transurethral resection of bladder tumors. Tissues were infected with $10^6$ PFU of the indicated viruses using buffered saline as a negative control (mock). The images from top to bottom represent a white light tissue image, mCherry signal, an overlay, and a heat map image showing mCherry expression, respectively (scale bars = 1 mm).

B   Quantification of the mCherry expression in panel (A) at 24 h post-infection. Mock-infected cells were used as background correction. Represent one primary tissue that was analyzed as described in Data Information.

C   *Ex vivo* infection of high-grade T2 (UCKP-4) bladder tumor as in (A).

D   Quantification of the mCherry expression in panel (C) at 24 h post-infection. Represent one primary tissue that was analyzed as described in Data Information..

E   Representative (4×) microscope pictures of the UCKP-4 (top) and UCKP-6 (bottom) primary cell cultures used in (F).

F   Growth of the indicated VACV strains in UCKP-4 primary human bladder cancer cultures. The cells were infected at 0.03 PFU/cell, harvested at the indicated times, and titered on BSC-40 cells. Data represent single lysates titered in duplicate.

Data information: Quantification in (B) and (D) shows the mean mCherry signal intensity (value per pixel) over the area of a given tumor sample. Data are shown as the mean ± SEM of these intensities over the tumor area.

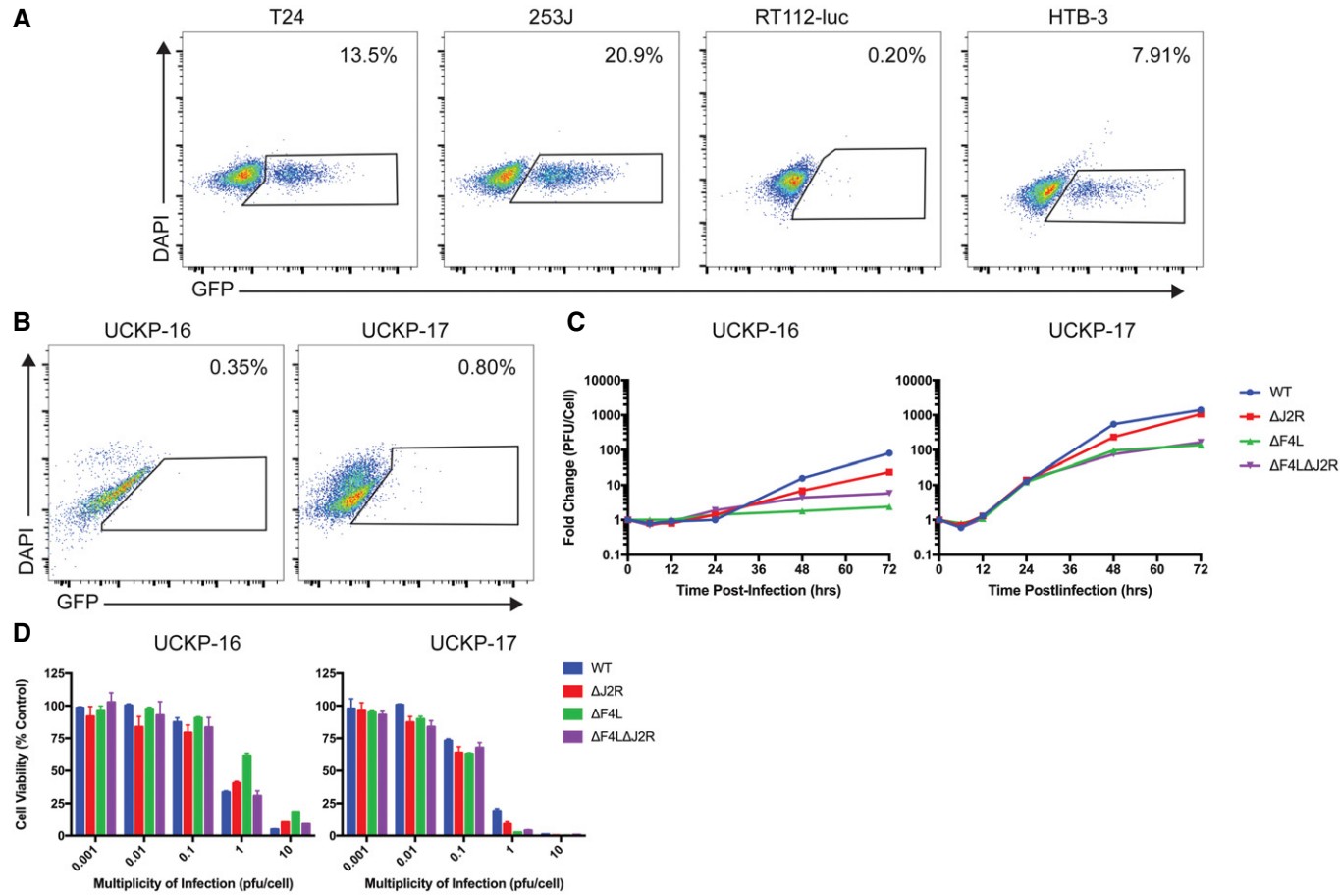

**Figure 8. VACV replicates in a BCG-resistant established cell line and in primary bladder tumor cultures.**

A   BCG uptake in different cell lines. Cells were incubated with BCG-GFP for 24 h, and then BCG uptake was measured by flow cytometry. The numbers show the percentage of GFP-positive events out of total events. DAPI was used as an empty channel and the gates were set based on uninfected cells.

B   BCG uptake by primary bladder tumor cultures. UCKP-16 represents a low-grade Ta tumor and UCKP-17 represents a low-grade T1 tumor. Cells were treated and analyzed as in (A).

C   Growth of the indicated VACV strains in subconfluent primary human bladder tumor cultures *in vitro*. Cells were infected at a multiplicity of infection of 0.03 PFU/cell, harvested at the indicated times, and titered on BSC-40 cells.

D   VACV killing of UCKP-16 and UCKP-17 cells. Subconfluent cells were infected at the indicated multiplicities of infection (PFU/cell), cultured for 3 days, and assayed for viability with resazurin dye. Uninfected cells were used as control.

Data Information: Data in (D) are shown as the mean ± SEM of single lysates titered in duplicate.

complementation of the viral RRM2 deletion, even under low serum conditions.

We observed robust replication of both ΔF4L VACV and ΔF4LΔJ2R VACV in most bladder cancer cell lines and in BCG-resistant RT112-luc cells. Additionally, we have shown that ΔF4LΔJ2R VACV can replicate in, and kill, BCG-resistant primary bladder cancer cultures. Unlike a *J2R*-deleted mutant, replication of the *F4L*-deleted viruses was attenuated in non-proliferating (partially serum-deprived) normal cells.

For our initial *in vivo* experiments, we used both subcutaneous and orthotopic RT112-luc tumors as models for NMIBC. This provides a replacement for HeLa-contaminated KU7 or KU7-luc cells (Jäger *et al*, 2013) and permitted bioluminescence monitoring of orthotopic tumor progression. RT112 cells have been used previously to model MIBC cancer by injecting cells into the bladder wall (Gust *et al*, 2013). However, MIBC is not treated by intravesical

therapies. Our cell instillation technique produces a model for NMIBC, for which intravesical therapies are appropriate. These orthotopic RT112-luc tumors responded dramatically to intravesical administration of the three mutant VACVs. In addition, intratumoral and intravenous injection of each of the three VACVs in the subcutaneous xenograft model produced tumor control in a manner that corresponded roughly to the degree of virus replication in the tumor. It is interesting to note that ΔF4LΔJ2R VACV consistently produced better anti-tumor activity than ΔF4L VACV in our animal models, and often *in vitro* as well. This seems counterintuitive, and the reason(s) why are unclear, but the mechanism is being investigated.

The VACV Western Reserve strain was originally adapted for growth in mice and exhibits virulence in this model. However, we saw no signs of toxicity in any of the immune-compromised mice treated with ΔF4L or ΔF4LΔJ2R VACV, while significant toxicity was

observed in animals treated with ΔJ2R VACV. In addition, ΔJ2R VACV was recovered from multiple normal organs after intratumoral or systemic treatments. These data show that the J2R mutation does not suffice to prevent VACV replication in normal tissues of immune-compromised mice. Although a number of oncolytic J2R-deleted VACV strains have been used safely in many clinical trials, including one based on the VACV strain WR (Zeh et al, 2015), a further improvement might be obtained by incorporating F4L mutations. [In this regard, it is notable that Pexa-Vec/JX-594, a VACV lacking J2R, produced pox lesions in some patients (Cripe et al, 2015; Kung et al, 2015; Park et al, 2015)]. In another pre-clinical study, Fend et al described an oncolytic WR strain bearing deletions in J2R and in I4L, which encodes the large subunit of ribonucleotide reductase (Fend et al, 2016). While this virus showed promise as a therapeutic agent, in our experience virus replication is regulated more stringently by F4L than by I4L, both in vitro and in mouse models (Gammon et al, 2010). This is likely due to RRM2 expression being cell-cycle-regulated whereas RRM1 is not (Gammon et al, 2010).

Our VACVs were also evaluated in an AY-27 immune-competent orthotopic rat model of bladder cancer. AY-27 tumors resemble high-grade human urothelial cell carcinoma (Xiao et al, 1999; Hendricksen et al, 2008) and an oncolytic reovirus has been previously tested in this model, where it proved more effective and less toxic than BCG (Hanel et al, 2004). In the current study, we saw that all rats treated with live VACV showed a reduction in tumor size, with complete tumor clearance in many animals, and a significant increase in survival relative to the controls. One caveat with the AY-27 model is that up to 30% of the tumors invade the muscle by day 6 (Xiao et al, 1999; Hendricksen et al, 2008) and this characteristic may explain why some animals have a limited response to these therapies. While all three live VACV treatments produced statistically similar outcomes, it is interesting to note that regular monitoring of tumors by cystoscope showed that some of the rats treated with ΔJ2R VACV had relapsed and this was not seen in rats treated with the F4L-deleted VACVs. The ΔJ2R VACV-treated animals had significantly higher levels of neutralizing antibodies suggestive of a more robust anti-viral, and thus more oncolysis-limiting, response.

Outside of the previously cited studies, there has been little other pre-clinical work with oncolytic viruses in immune-competent bladder cancer models. Fodor et al studied a p53-expressing J2R-deleted VACV in an orthotopic MB49 immune-competent mouse model and obtained three out of nine long-term survivors (Fodor et al, 2005). OncoVEX^GALV/CD, a herpes simplex virus type 1 engineered to express cytosine deaminase and a fusogenic glycoprotein, was tested in the AY-27 rat bladder cancer model (Simpson et al, 2012) and caused a significant reduction in tumor volume. Unfortunately, in neither study was there any further investigation to establish whether treatment generated an anti-tumor immune response.

As in the xenograft studies, the two F4L-deleted VACVs proved highly tumor-selective in the rat AY-27 model. In contrast, ΔJ2R VACV was detected in the lungs, kidneys, and ovaries in a subset of animals, although spread was not associated with overt toxicity. The ovaries consistently had the second highest levels of ΔJ2R VACV after the tumor and we also saw what appeared to be ovarian cysts in multiple mice and rats. Other J2R-deleted VACVs are reported to replicate in normal mouse tissues,

including the ovaries where they can cause pathology and sterility (MacTavish et al, 2010; Zhao et al, 2011; Gentschev et al, 2012).

All rats that remained tumor-free through day 125 (as determined by cystoscopy) exhibited anti-tumor immunity as shown by tumor rejection upon challenge. In vitro assays performed 100 days after challenge confirmed the presence of long-lasting tumor-specific CD8^+ T cells in all cured ΔF4LΔJ2R-treated rats. It is notable that these tumor-specific cytotoxic T-lymphocytes were only detected in animals that had also been exposed to live virus. Surprisingly, we know of no clear evidence that BCG can induce a protective anti-tumor immune response (Ratliff et al, 1993; Gan et al, 1999). If BCG treatment is not also generating anti-tumor immunity, it could explain the high recurrence rate in BCG-treated patients.

As a final experiment, we showed that primary human bladder cancer tissues, in the form of either monolayers or as tumor fragments, could support the replication of both ΔJ2R and ΔF4LΔJ2R VACV ex vivo to a much greater degree than was seen in normal urothelium. This effect is similar to what has been reported for Pexa-Vec (JX-594) using rectal, endometrial, and colon cancer fragments along with adjacent normal tissues (Breitbach et al, 2011). The fact that ΔF4LΔJ2R VACV also grew selectively in primary explanted human bladder cancer tissues provides some promise that these pre-clinical results might also be seen in patients treated with oncolytic VACV.

In conclusion, these results suggest that NMIBC could be a highly suitable target for oncolytic treatment with a WR-based ΔF4LΔJ2R VACV and we are currently planning a phase I/II clinical trial with a similar VACV construct. From a practical perspective, intravesical bladder delivery offers a way of delivering high doses of virus directly to the tumor while also helping to limit systemic spread. ΔF4LΔJ2R VACV showed an impressive safety profile, selectively infected a variety of susceptible cell types (including primary human bladder cancer tissues), and infection induced anti-tumor immunity. Although patients with immune deficiencies and BCG-refractory cancers would be ideal candidates for this therapy, in the longer-term oncolytic ΔF4LΔJ2R VACV might offer a more attractive replacement for BCG and potentially reduce the need for surgical management.

## Materials and Methods

### Cell lines

The human (253J, RT4-luc, HTB-3, HTB-9, MGH-U3, RT112-luc, T24) and rat (AY-27) bladder cancer cell lines were maintained in RPMI 1640 medium supplemented with 10% FBS, 2 mM L-glutamine, 100 U/ml penicillin, 100 U/ml streptomycin, and 0.25 μg/ml Fungizone^® (Gibco). The HT-1376, UM-UC3-luc, UM-UC6, UM-UC9, and UM-UC14-luc human bladder cancer cell lines were maintained in Dulbecco's modified Eagle's medium (DMEM)/F12 medium with the same supplements. MB49-luc (murine urothelial cell carcinoma), N60 (early passage human skin fibroblast), NKC (normal human kidney epithelial), HeLa (CCL-2), and RK3E (E1A immortalized rat kidney) cells were cultured in DMEM, also with the same supplements. BSC-40 cells (CRL-2761) were grown in minimal

essential medium (MEM) supplemented as above, but using 5% FBS. Cells were cultured at 37°C in 5% $CO_2$. RT4-luc, RT112-luc, and UM-UC3-luc cell lines were kindly provided by D. McConkey (MD Anderson), MB49 cells were kindly provided by J. Greiner (National Cancer Institute), and early passage N60 human skin fibroblast cells were kindly provided by T. Tredget (University of Alberta). All lines tested free of mycoplasma either by Hoechst 33342 staining (Thermo Fisher Scientific) and fluorescence imaging, or using a LookOut® Mycoplasma PCR detection kit (Sigma-Aldrich). The identities of the cell lines were confirmed using a 16-marker AmpFLSTR® Identifiler® system and tests performed by the TCAG facility at the University of Toronto.

## Primary cell culture

Primary cancer and adjacent normal tissues were obtained from consenting patients undergoing surgery with the approval of the University of Alberta Health Research Ethics Board. Samples were received in saline and processed within 2 h of surgery. Submucosal and necrotic tissues were stripped from the tumor tissue and the remaining tumor was processed in 4 ml of "spleen dissociation medium" (STEMCELL Technologies) using a GentleMACS dissociator (Miltenyi Biotec). After rocking at 37°C for 30 min, the suspension was reprocessed, EDTA was added to 10 mM, and the cells were incubated at room temperature for 10 min. The cells were centrifuged for 5 min at 350 g, washed, and plated in EpiLife medium (Thermo Fisher Scientific) supplemented with 25 μg/ml bovine pituitary extract, 0.5 ng/ml epidermal growth factor, 3 mM glycine, 0.1 mM MEM non-essential amino acids, 1% ITS (insulin, transferrin, selenium), 2% FBS, 2 mM L-glutamine, 100 U/ml penicillin, 100 U/ml streptomycin, and 0.25 μg/ml Fungizone® (Gibco).

## Biological agents

All recombinant viruses used in this study were derived from a clonal isolate of VACV strain WR, originally obtained from the American Type Culture Collection (ATCC). To permit in vivo imaging, we generated new versions of the ΔF4L and ΔJ2R viruses described in Gammon et al (2010) that encode mCherry fluorescent protein under virus early/late promoter control. Detailed construction of the viruses is described in the Appendix Supplementary Methods.

GFP-expressing BCG (BCG-GFP) was a generous gift of Drs. Gil Redelman-Sidi and Michael Glickman (MSKCC) and has been previously described (Redelman-Sidi et al, 2013). Briefly, the BCG Pasteur strain was transformed with pYUB921 (episomal plasmid encoding GFP and conferring kanamycin resistance).

## In vitro infection experiments

Multistep growth curves were obtained by infecting the indicated cell lines with VACV at MOI of 0.03 PFU/cell. Infected cells were scraped into the culture medium at the indicated times, subjected to three rounds of freeze thaw and then diluted and plated in duplicate on BSC-40 cells. Infected BSC-40 cells were cultured in medium containing 1% carboxymethyl cellulose for 2 days and then fixed and stained with crystal violet. Plaque counts were determined from wells containing 30–250 plaques.

## Cytotoxicity assays

Cells were seeded in 48-well plates at a density estimated to produce ~50% confluency at the time of infection. After an 8-h incubation, cells were infected with VACV, cultured for 3 days and the media replaced with fresh cell culture media containing 44 μM resazurin (Sigma-Aldrich). Plates were incubated 4–6 h at 37°C, and then fluorescence was read using a FLUOstar plate reader (BMG Labtech) with 560-nm excitation/590-nm emission filters.

## In vivo tumor models

All studies reported in this communication were conducted with the approval of the University of Alberta Health Sciences Animal Care and Use Committee in accordance with guidelines from the Canadian Council for Animal Care. Animals were housed with access to food and water ad libitum in ventilated mouse or rat cages (1–5 mice per cage, 1–2 rats per cage) in a biosafety level 2 containment suite at the University of Alberta Health Sciences Laboratory Animal Services Facility.

For all xenograft models, female Balb/c nude mice (Charles River Laboratories) were 8 weeks old and at least 16 g in weight at the time of tumor implantation. To establish orthotopic RT112-luc tumors, mice were anesthetized with 2% isoflurane and a 27G angiocatheter (BD Biosciences), with the needle removed, was lubricated with sterile Lubrifax and inserted into the bladder via the urethra. The bladder was infused for 15 s with 50 μl of 0.1 M HCl, neutralized with 50 μl of 0.1 M KOH for 15 s, and then washed three times with phosphate-buffered saline (PBS). Next, a 50 μl suspension of Hank's balanced salt solution (HBSS) containing $2 \times 10^6$ RT112-luc cells was instilled into the bladder using the catheter and left in-dwell for 1 h while the mice remained under anesthesia. For virus treatments, the bladders of anesthetized mice were emptied by catheterization and then 50 μl of PBS, containing $1 \times 10^6$ PFU of virus, was instilled into each mouse on days 10, 13, and 16 post-implantations. The virus was left in-dwell for 1 h while the mice remained under anesthesia.

To produce RT112-luc or UM-UC3-luc flank tumors, Balb/c nude mice were anesthetized with isoflurane then injected subcutaneously with 0.1 ml of $2 \times 10^6$ tumor cells in PBS containing 50% Matrigel (Corning).

Ten-week-old Fisher F344 immune-competent female rats (Charles River Laboratories), weighing at least 150 g, were used for orthotopic AY-27 tumor implantation as previously described (Xiao et al, 1999). Briefly, 0.3 ml 0.1 M HCl was instilled into the bladder of rats anesthetized with isoflurane, left in-dwell for 15 s, and neutralized with 0.3 ml of 0.1 M KOH for 15 s, and then the bladder washed three times with PBS. The catheter was then used to deliver 0.3 ml of saline containing $3 \times 10^6$ AY-27 cells, the cells were left in-dwell for 1 h, and the rats were returned to their cages. Five days later, tumor take was confirmed by cystoscopy (Asanuma et al, 2003). For virus treatments, the rats were anesthetized and catheterized, the bladders were emptied by suprapubic pressure, and then $3 \times 10^8$ PFU of virus in 0.3 ml PBS was instilled into each bladder on days 6, 9, and 12 and left in-dwell for 1 h.

Rats that were determined to be tumor-free by cystoscopy at day 125 post-tumor implantations were challenged with $3 \times 10^6$ AY-27

cells in the flank. Cells were resuspended in HBSS and mixed with equal volumes of Matrigel (Corning). A total of 200 μl was injected. Age-matched Fisher F344 immune-competent female rats were used as controls.

## Isolation of CD3+ cells

Spleens were harvested from euthanized rats and placed in HBSS on ice. Next, they were cut into small pieces, resuspended in 4 ml of spleen dissociation medium, and broken up using a GentleMACS dissociator, followed by rocking at 37°C for 30 min. The dissociation program was run again, EDTA was added to 10 mM, and the cells were incubated at room temperature for another 10 min. Cells were then filtered through a 70-μm MACS SmartStrainer (Miltenyi Biotec), centrifuged for 5 min at 350 g, and washed with HBSS. Red blood cells were lysed, and then the remaining cells were recovered by centrifugation, then resuspended in PBS and counted. CD3+ cells were isolated from this preparation using a MagCellect™ Rat CD3+ T-cell isolation kit following the manufacturer's protocol (R&D Systems).

## BMDC culture and lysate loading

The femurs were removed from euthanized naïve Fischer F344 female rats, cleaned of attached tissue, soaked in 70% isopropanol for 2 min, and rinsed in HBSS. Femur ends were removed and the marrow was flushed with HBSS. Red blood cells were lysed (eBioscience), and $3 \times 10^6$ of the remaining bone marrow cells were plated on 100-mm untreated plates in 8 ml RPMI 1640 medium supplemented with 10% FBS, 50 μM 2-mercaptoethanol, 2 mM L-glutamine, 100 U/ml penicillin, 100 U/ml streptomycin, 0.25 μg/ml Fungizone® (Gibco), 500 U/ml rat GM-CSF (Peprotech), and 20 ng/ml rat IL-4 (Peprotech). The cells were cultured at 37°C and the medium, still containing GM-CSF and IL-4, was replaced on day 3. On day 5, the medium was replaced again, and 100 μg/ml of AY-27 tumor lysate was added (see Appendix Supplementary Methods for preparation of tumor lysate). 12 h later, the cultures were matured by adding 20 U/ml TNF-α (Peprotech) and 0.5 μg/ml CD40L (AdipoGen) (Labeur et al, 1999; Bachleitner-Hofmann et al, 2002). Two days after adding the tumor cell lysates, the cultures were resuspended at $1 \times 10^6$ cells/ml in fresh medium.

## T-lymphocyte assays

Proliferation assays were performed in 96-well U-bottom plates (Greiner Bio-One). BMDCs (pulsed with or without lysates) were co-cultured with $10^5$ CD3+ cells at different ratios (1:1, 10:1, and 100:1 CD3:BMDC) in RPMI 1640 medium as described above. The CD3+ cells were previously labeled with CellTrace Violet per the manufacturer's directions (Thermo Fisher Scientific). After 6 days of co-culture, flow cytometry was used to measure CD3+ T-cell proliferation. Supernatants were collected from cells co-cultured for 24 h, and assayed by ELISA for interferon-γ (Legend Max, BioLegend™).

Cytotoxicity assays were performed using $10^5$ rat splenic CD3+ cells co-cultured in 96-well U-bottom plates with $10^4$ BMDCs in RPMI 1640, supplemented as described above. On day 7, the CD3+ cells were collected and incubated for 18 h in flat-bottom 96-well

### The paper explained

#### Problem

Up to 80% of non-muscle-invasive bladder cancers (NMIBC) can recur within 5 years of initial treatment, with high-grade NMIBC posing the greatest risk of recurrence. Treatment for these patients includes transurethral resection followed by intravesical therapy with the immunotherapeutic agent Bacillus Calmette–Guérin (BCG). BCG, however, carries the risk of systemic infection and can be particularly dangerous for immunocompromised patients. Additionally, up to 40% of patients fail BCG therapy and cystectomy remains the standard treatment in these cases. There is an urgent need for more bladder-sparing therapies for patients failing conventional therapies.

#### Results

Here, we report the generation and testing of a novel oncolytic VACV that is lacking the viral F4L and J2R genes, homologs of cellular genes RRM2 and TK, respectively, that promote nucleotide biosynthesis in infected cells. This oncolytic virus, ΔF4LΔJ2R VACV, is highly attenuated in non-cancerous cells and replicated selectively in both an orthotopic AY-27 immunocompetent rat tumor model and an RT112-luc xenografted human tumor model, causing significant tumor regression or complete tumor ablation with no toxicity. In contrast, a VACV with a more commonly employed deletion, ΔJ2R, spread to normal organs and caused significant toxicity in immunocompromised mice. Furthermore, rats cured of AY-27 tumors by VACV treatment developed a protective anti-tumor immunity that was evidenced by tumor rejection upon challenge, as well as by ex vivo cytotoxic T-lymphocyte assays. Finally, ΔF4LΔJ2R VACV replicated in an established BCG-resistant human bladder cancer cell line and in fresh cultures of primary human bladder tumors.

#### Impact

This study demonstrated the high degree of safety and anti-tumor activity of a novel oncolytic virus in pre-clinical bladder cancer models. Given the high rate of recurrence and the lack of treatment options for BCG-resistant bladder cancer, our oncolytic VACV could provide a safe and urgently needed therapy for BCG failure.

plates, along with target cells, at effector-to-target ratios ranging from 20:1 to 0.625:1. The plates were assayed for lysis by LDH assay (Thermo Scientific Pierce). CD107a expression was measured essentially as described by Betts et al (2003). BMDCs were added to CD3+ cells, co-cultured for 1 h in the presence of CD107a antibody, and incubated for another 5 h in the presence of monensin and brefeldin A (BioLegend). The cells were fixed and stained with anti-CD4, anti-CD8, and secondary to CD107a antibodies and analyzed by flow cytometry.

## Statistics

Data were analyzed using two-tailed Student's t-test when comparing the means of two groups. Multiple t-test was used to determine significance of VACV growth after siRNA knockdown. Analysis of variance (ANOVA) was used when comparing multiple groups followed by Tukey's multiple comparison test. Microarray data were analyzed in the RStudio programming environment (v0.98.501). Significance analysis was performed by means of a one-way ANOVA followed by Tukey's HSD. Data for animal survival curves were analyzed by log-rank (Mantel–Cox) test. The numbers of animals included in each figure are indicated at the end of each legend. P-values are indicated within each figure.

**Expanded View** for this article is available online.

## Acknowledgements

We thank Ted Tredget (University of Alberta), David McConkey (MD Anderson), John Greiner (National Cancer Institute), and Michael Glickman (MSKCC) for reagents. Flow cytometry was performed at the University of Alberta, Faculty of Medicine and Dentistry Flow Cytometry Facility, with assistance from Aja Rieger. We would also like to thank Powel Crosley and Nubia Zepeda for comments on the manuscript as well as David Sharon, Shyambabu Chaurasiya, and Ryan Noyce for helpful discussions. Finally, we thank the patients for providing primary tissue. This work was supported by an Alberta Cancer Foundation/Canadian Institutes for Health Research (CIHR) Bridge award #BOG 25964, Alberta Innovates Health Solutions (AIHS)/Alberta Cancer Foundation "High Risk for High Return" award #201000559, a Li Ka Shing Institute of Virology "Translational Research" grant, and a CIHR grant MOP 130473 [to D.H.E., M.M.H., and R.B.M.]. Funding from the AIHS in support of the Li Ka Shing Institute of Virology and Applied Virology [to Drs. L. Tyrrell, M. Houghton, and D.H.E.] is also acknowledged. K.G.P. stipend was in part funded by the Antoine Noujaim Graduate Entrance Scholarship, Queen Elizabeth II Graduate Scholarship, and the University of Alberta Faculty of Medicine and Dentistry 75th Anniversary Graduate Student Award.

## Author contributions

KGP designed and performed experiments, analyzed the data, and prepared the manuscript. CRI designed the viruses used in these studies, performed the siRNA silencing experiment, and contributed to experimental design. NAF assisted in all animal experiments. RBM collected patient specimens and DBP and JDL performed *ex vivo* tumor imaging experiments and analyzed resulting data. KMV assisted in microarray data analysis. DHE, MMH, and RBM provided guidance in experimental design, data interpretation, and manuscript preparation.

## Conflict of interest

Dr. Evans has been awarded U.S. patents as a co-inventor of related oncolytic virus technologies and is a co-owner of Prophysis Inc., which retains a partial interest in the licensing rights for these technologies. The other authors declare that they have no conflict of interest.

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
