## [Review Process File · EMBO Molecular Medicine]

Deletion of F4L (ribonucleotide reductase) in vaccinia virus produces a selective oncolytic virus and promotes anti-tumor immunity with superior safety in bladder cancer models

Kyle G. Potts, Chad R. Irwin, Nicole A. Favis, Desmond B. Pink, Krista M. Vincent, John D. Lewis, Ronald B. Moore, Mary M. Hitt, David H. Evans

Corresponding author: David Evans, University of Alberta

Review timeline:

Submission date:	04 November 2016
Editorial Decision:	24 November 2016
Revision received:	26 January 2017
Editorial Decision:	09 February 2017
Revision received:	10 February 2017
Accepted:	15 February 2017

Transaction Report:

(Please note that the manuscript was previously reviewed at another journal and the reports were taken into account in the decision making process at EMBO Molecular Medicine. Since the original reviews are not subject to EMBO's transparent review process policy, the reports and author response cannot be published.)

Editor: Roberto Buccione

1st Editorial Decision

24 November 2016

We have now heard from the expert external advisor whom we asked to help us on making a decision on your manuscript.

The advisor was provided with the full manuscript, the previous Reviewers' comments and your Point-by-Point rebuttal.

As you will see, s/he agrees that the manuscript would make an interesting and worthy contribution, pending an experimental concern and a number of issues that need to be addressed. After internal discussion we agreed that, provided you successfully address these concerns, we would be happy to proceed with your manuscript without further revision other than going back to the advisor for a final check, if necessary.

I look forward to seeing a revised form of your manuscript as soon as possible.

***** Reviewer's comments *****

Referee #1 (Remarks):

I have now had chance to look at the paper in more detail. When the key finding of the paper is

clear, I think it does carry significance for the field. Hence my overall recommendation is to accept it with a few amendments - including sharpening of the title and abstract to convey the key findings more clearly.

In my view the most important finding of this manuscript is that deletion of the RRM2 gene homologue from vaccinia virus gives it cancer selectivity under low serum conditions (used to mimic conditions within a tumour environment). The immune effects are consequential to this, but are not the key aspect. Hence I think the title should be modified to something such as: "Deletion of F4L (ribonucleotide reductase) in vaccinia virus produces a selective oncolytic phenotype and promotes anti-tumor immunity with superior safety in bladder cancer models". I would suggest the Abstract is also sharpened to make the key finding obvious.

1. It is not very clear to me why N60 fibroblasts were used as a 'normal' cell control, when the bladder cancer cell lines are epithelial - why not use an immortalised normal epithelial cell such as Human Airway Epithelial Cells etc?
2. They imply that low serum acts at least partly through effects on the cell cycle; is there data somewhere showing that the low serum conditions does stall the cell cycle in those cells? Is it feasible that the only difference between the cancer cells and the normal fibroblasts is that the former were not able to quiesce, and this explains the effects on the F4L-deleted viruses? That does not really undermine the interpretation, although it is important and should definitely be discussed.
3. Also the Discussion should contain a paragraph discussing what metabolic deregulation may have occurred in the cancer cell lines that allows complementation of the RRM2 deletion even under low serum conditions.

It is notable that the majority of comments from the previous referees have focused on the immunostimulatory properties of the viruses, which presumably just reflects the different patterns of virus activity in vivo. Personally I think the animal data are sufficiently convincing, but the authors need to pay more attention to the cellular mechanisms that govern the oncolytic activity observed - as outlined above.

1st Revision - authors' response

26 January 2017

Referee #1 (Remarks):

I have now had chance to look at the paper in more detail. When the key finding of the paper is clear, I think it does carry significance for the field. Hence my overall recommendation is to accept it with a few amendments - including sharpening of the title and abstract to convey the key findings more clearly.

In my view the most important finding of this manuscript is that deletion of the RRM2 gene homologue from vaccinia virus gives it cancer selectivity under low serum conditions (used to mimic conditions within a tumour environment). The immune effects are consequential to this, but are not the key aspect. Hence I think the title should be modified to something such as: "Deletion of F4L (ribonucleotide reductase) in vaccinia virus produces a selective oncolytic phenotype and promotes anti-tumor immunity with superior safety in bladder cancer models". I would suggest the Abstract is also sharpened to make the key finding obvious.

The title has been modified based on the reviewer's suggestions. We have also made some revisions to the abstract.

1. It is not very clear to me why N60 fibroblasts were used as a 'normal' cell control, when the bladder cancer cell lines are epithelial - why not use an immortalised normal epithelial cell such as Human Airway Epithelial Cells etc?

We have added a normal epithelial kidney (NKC) cell line to our growth curve and cytotoxicity panels (Figures EV2 and EV3). The results are similar to what we observed with N60 fibroblasts.

2. They imply that low serum acts at least partly through effects on the cell cycle; is there data somewhere showing that the low serum conditions does stall the cell cycle in those cells? Is it feasible that the only difference between the cancer cells and the normal fibroblasts is that the former were not able to quiesce, and this explains the effects on the F4L-deleted viruses? That does not really undermine the interpretation, although it is important and should definitely be discussed.

We have analyzed distribution of human RT112-luc bladder cancer cells under both regular and low serum conditions. We have also carried out cell cycle analysis on the N60 cell line and the NKC cell line under the same conditions (Figure EV4). This is discussed in the text (Page 6).

3. Also the Discussion should contain a paragraph discussing what metabolic deregulation may have occurred in the cancer cell lines that allows complementation of the RRM2 deletion even under low serum conditions.

A paragraph has been added to the discussion to address the metabolic deregulation that may occur leading to complementation of Δ F4L VACV (Page 15).

It is notable that the majority of comments from the previous referees have focused on the immunostimulatory properties of the viruses, which presumably just reflects the different patterns of virus activity in vivo. Personally I think the animal data are sufficiently convincing, but the authors need to pay more attention to the cellular mechanisms that govern the oncolytic activity observed - as outlined above.

2nd Editorial Decision

09 February 2017

Thank you for the submission of your revised manuscript to EMBO Molecular Medicine. We have now received the enclosed report from the reviewer who was asked to re-assess it. As you will see the s/he is now globally supportive and I am pleased to inform you that we will be able to accept your manuscript pending the following final minor amendments:

- 1) Please provide a running title in the title page
- 2) We noticed the figure callout for Appendix Figure S5A-D on page 10, which is most likely a mistake. Did you mean perhaps Appendix Figure S6A-D? Also, and probably connected to this, you mention Appendix Figure S6A and B but not C and D.
- 3) Please remove the Appendix methods from the main text
- 4) The scale bars for Fig. 7A are very difficult to see, please improve
- 5) We noted that in Fig. 7A, the mCherry heat Map mock panel appears to be empty. Could you please verify and eventually clarify?

Please submit your revised manuscript within two weeks. I look forward to seeing a revised form of your manuscript as soon as possible so that we can proceed with formal acceptance.

***** Reviewer's comments *****

Referee #1 (Comments on Novelty/Model System):

The paper has been revised to address my previous concerns and now contains a clearer focus on the dependence of the virus on tumour metabolism alongside their observations about immunogenicity and creation of an anticancer immune response. The work now fits well into the emerging field of oncolytic viruses and should be well cited. All my concerns have been adequately addressed.

Thank you for the submission of your revised manuscript to EMBO Molecular Medicine. We have now received the enclosed report from the reviewer who was asked to re-assess it. As you will see the s/he is now globally supportive and I am pleased to inform you that we will be able to accept your manuscript pending the following final minor amendments:

1) Please provide a running title in the title page

We have added the following running title “ Δ F4L oncolytic VACV in bladder cancer”

2) We noticed the figure callout for Appendix Figure S5A-D on page 10, which is most likely a mistake. Did you mean perhaps Appendix Figure S6A-D? Also, and probably connected to this, you mention Appendix Figure S6A and B but not C and D.

Errors have been corrected.

3) Please remove the Appendix methods from the main text

Appendix text has been removed from the end of the manuscript.

4) The scale bars for Fig. 7A are very difficult to see, please improve

We have increased the size and clarity of the scale bars.

5) We noted that in Fig. 7A, the mCherry heat Map mock panel appears to be empty. Could you please verify and eventually clarify?

mCherry signal in the mock panel is correct. There is no visible signal generated from the mock infected tissue.

Please submit your revised manuscript within two weeks. I look forward to seeing a revised form of your manuscript as soon as possible so that we can proceed with formal acceptance.

We also made a formatting correction to the Appendix document as well.

Corresponding Author Name: DAVID EVANS

Journal Submitted to: EMBO MOLECULAR MEDICINE

Manuscript Number: EMM-2016-07296